# Graph Hawkes Neural Network for Forecasting on Temporal Knowledge Graphs

**Zhen Han**                                                      ZHEN.HAN@CAMPUS.LMU.DE
**Yunpu Ma**[*]                                             COGNITIVE.YUNPU@GMAIL.COM
*LMU Munich & Siemens AG*
*Otto-Hahn-Ring 6, 81739 Munich, Germany*

**Yuyi Wang**                                                          YUWANG@ETHZ.CH
*ETH Zürich*
*Rämistrasse 101, 8092 Zürich, Switzerland*

**Stephan Günnemann**                                          GUENNEMANN@IN.TUM.DE
*Technical University of Munich*
*Boltzmannstr. 3, 85748 Garching b. München, Germany*

**Volker Tresp**[*]                                          VOLKER.TRESP@SIEMENS.COM
*LMU Munich & Siemens AG*
*Otto-Hahn-Ring 6, 81739 Munich, Germany*

## Abstract

The Hawkes process has become a standard method for modeling self-exciting event sequences with different event types. A recent work has generalized the Hawkes process to a neurally self-modulating multivariate point process, which enables the capturing of more complex and realistic impacts of past events on future events. However, this approach is limited by the number of possible event types, making it impossible to model the dynamics of evolving graph sequences, where each possible link between two nodes can be considered as an event type. The number of event types increases even further when links are directional and labeled. To address this issue, we propose the Graph Hawkes Neural Network that can capture the dynamics of evolving graph sequences and can predict the occurrence of a fact in a future time instance. Extensive experiments on large-scale temporal multi-relational databases, such as temporal knowledge graphs, demonstrate the effectiveness of our approach.

## 1. Introduction

If political relations between two countries becomes more tense, will it affect the international trades between them? If yes, which industries will bear the brunt? Modeling the relevant events that can be temporarily affected by international relations is the key to answer this question. However, the issue of how to model these complicated temporal events is an intriguing question. A possible way is to embed events in a temporal knowledge graph, which is a graph-structured multi-relational database that stores an event in the form of a quadruple. Events are point processes and point process models, in the past, have been widely applied to many real-world applications such as the analysis of social networks [Zhou et al., 2013], the prediction of recurrent user behaviors [Du et al., 2016], and the estimation

---

*. Corresponding author

of consumer behaviors in finance [Bacry et al., 2016]. The well known **Poisson process** [Palm, 1943] is limited to modeling temporal events that occur independently of one another. Hawkes [1971] proposed a self-exciting point process, which is now known as the **Hawkes process**, which assumes that past events have an excitation effect on the likelihood of future events, and such excitation exponentially decays with time. This model has been shown to be effective in modeling earthquakes [Ogata, 1998]. However, it is unable to capture some real-world patterns where past events of a different type may have inhibitory effects on future events, i.e., a skateboard purchase may inhibit a bike purchase. To address this limitation, the neural Hawkes process [Mei and Eisner, 2017] generalized the Hawkes process using recurrent neural networks with continuous state spaces such that past events can excite and inhibit future events in a complex and realistic way. Nevertheless, the neural Hawkes process is only capable of modeling event sequences with a small number of event types and fails to accurately capture the mutual influence in large-scale temporal multi-relational data. An example would be the evolving links in a dynamic graph sequence where the connections between nodes can be considered as different event types. The problem becomes even more challenging when the links are directional and labeled. In order to model the dynamics of directional and labeled links in a graph sequence, we develop a novel Graph Hawkes Process and apply it to large-scale temporal multi-relational databases, such as temporal knowledge graphs.

Before introducing temporal knowledge graphs, we briefly review semantic knowledge graphs (semantic KGs), which are multi-relational knowledge bases for storing factual information. Semantic KGs such as the Google Knowledge Graph [Singhal, 2012] represent an event as a semantic triplet$(s, p, o)$ in which $s$ (subject) and $o$ (object) are entities (nodes), and $p$ (predicate) is a directional labeled link (edge). Latent feature models [Ma et al., 2018a, Nickel et al., 2011] and graph feature models [Minervini et al., 2014, Liu and Lü, 2010] are two popular approaches to develop statistical models for semantic KGs. However, in contrast to static multi-relational data in semantic KGs, relations between entities in many real-world scenarios are not fixed and may change over time. Such temporal events can be represented as a quadruple $(s, p, o, t)$ by extending the semantic triplet with a time instance $t$ describing when these events occurred. Further an event may last for a period of time. For example, (John, lives in, Vancouver) could be true for many time steps, and (Alice, knows, John) might be true always. We can simply discretize such an event into a sequence of time-stamped events to store it in the form of quadruples. Appendix A shows an example of a temporal KG. By considering time, the semantic KGs are augmented into temporal knowledge graphs (tKGs), which creates the need for statistical learning that can capture dynamic relations between entities in tKGs. Modeling dynamic relations between entities over tKGs becomes more challenging than normal event streams since the number of event types is of order $N_e^2 \cdot N_p$, where $N_e$ and $N_p$ are the number of entities and predicates respectively. Recent studies on tKGs reasoning focused on augmenting entity embeddings with time-dependent components in a low-dimensional space [Kazemi et al., 2019, Sankar et al., 2018]. However, the existing temporal KG models either lack a principled way to predict the occurrence time of future events or ignore the concurrent facts within the same time slice.

In this paper, we propose a novel deep learning architecture to capture temporal dependencies on tKGs, called **Graph Hawkes Neural Network** (GHNN). More specifically, our main contributions are:

- We propose a Graph Hawkes Neural Network for predicting future events on large-scale tKGs. This is the first work that uses the Hawkes process to interpret and capture the underlying temporal dynamics of tKGs.
- Different from the previous tKG models with discrete state spaces, we model the occurrence probability of an event in continuous time. In this way, our model can compute the probability of an event at an arbitrary timestamp, which considerably enhances model's flexibility.
- We analyze previous problematic evaluation metrics and propose a new ranking metric for link prediction on temporal knowledge graphs.
- Compared to state-of-the-art time prediction models on tKGs, our approach can achieve more accurate results.

## 2. Background and Related Work

### 2.1 The Hawkes Process

The Hawkes process is a stochastic process for modeling sequential discrete events occurring in continuous time where the time intervals between neighboring events may not be identical. Moreover, the Hawkes process supposes that past events can temporarily excite future events, which is characterized via the intensity function. The intensity function $\lambda_k(t)$ represents the expected number of events with type $k$ in the interval of unit length. Thus, according to the survival analysis theory [Aalen et al., 2008], the density function that an event with the type $k$ occurs at $t_i$ is defined as

$$p_k(t_i) = \lambda_k(t_i) \exp(-\int_{t_L}^{t_i} \sum_k \lambda_k(s)ds), \tag{1}$$

where $t_L$ denotes the latest occurrence of any event without regarding its event type.

### 2.2 Future Prediction on Temporal Knowledge Graphs

Temporal knowledge graphs are multi-relational, directed graphs with labeled timestamped edges (predicates) between nodes (entities). Each timestamped edge represents a specific event that is formed by a predicate edge $p$ between a subject entity $s$ and an object entity $o$ at a timestamp $t$ and is denoted by a quadruple $(e_s, e_p, e_o, t)$, where $e_s, e_o \in \{1, ..., N_e\}$, $e_p \in \{1, ..., N_p\}$, $t \in \mathbb{R}^+$. A tKG can therefore be represented as an ordered sequence of quadruples, $\mathbb{E} = \{e_i = (e_{s_i}, e_{p_i}, e_{o_i}, t_i)\}_{i=1}^N$, where $0 \leq t_1 \leq ... \leq t_n$. A classic task in tKGs is to predict either a missing subject entity $(?, e_{p_i}, e_{o_i}, t_i)$ or a missing object entity $(e_{s_i}, e_{p_i}, ?, t_i)$. While one aims to predict the missing links in the existing graphs in the context of a semantic knowledge graph, one wants to predict the future links at a future timestamp $t_i$ based on observed events that occurred before $t_i$. Besides predicting what will happen in the future, another challenging problem is to predict when an event will happen, which is referred as the time prediction task. More concretely, one can precisely answer questions like:

- **Object prediction.** Which country will Emmanuel Macron visit next?

- **Subject prediction.** Who is the wife of Emmanuel Macron?

- **Time prediction.** When will Emmanuel Macron tweet again?

Recently, several studies focussed on temporal knowledge graph reasoning. Esteban et al. [2016] introduced an event model for modeling the temporal evolution of KGs where the prediction of future events is based on the latent representations of the knowledge graph tensor and of the time-specific representations from the observed event tensor. Jiang et al. [2016] augmented existing static knowledge graph models with temporal consistency constraints such as temporal order information and formulated the time-aware inference as an Integer Linear Program problem. In addition, Ma et al. [2018b] developed extensions of static knowledge graph models by adding a timestamp embedding to their score functions. Besides, Leblay and Chekol [2018] incorporated time presentations into score functions of several static KG models such as TransE [Bordes et al., 2013] and RESCAL [Nickel et al., 2011] in different ways. Additionally, García-Durán et al. [2018] suggested a straight forward extension of some existing static knowledge graph models that utilize a recurrent neural network (RNN) to encode predicates with temporal tokens derived by decomposing given timestamps. However, these models cannot generalize to unseen timestamps because they only learn embeddings for observed timestamps. In contrast, **LiTSEE** [Xu et al., 2019] directly incorporates time as a scale into entity representations by utilizing the linear time series decomposition. Also, **Know-Evolve** [Trivedi et al., 2017] learns evolving entity representations using the Rayleigh process, being able to capture the dynamic characteristics of tKGs. Additionally, **RE-Net** [Jin et al., 2019] augmented the **R-GCN** model [Schlichtkrull et al., 2018] to tKGs and uses the order of history event for predicting the future.

## 3. Notation

Throughout the following sections, $e_i$ denotes an event consisting of $(e_{s_i}, e_{p_i}, e_{o_i})$ where $e_{s_i}$, $e_{o_i}$ and $e_{p_i}$ written not in bold represent the subject entity, object entity and predicate of the event $e_i$, respectively. Additionally, we use $t_i$ to denote the timestamp when the event $e_i$ occurred. Besides, $\mathbf{e}_{s_i}$, $\mathbf{e}_{p_i}$, $\mathbf{e}_{o_i}$ written in bold represent their embeddings. We denote vectors by bold lowercase letters, such as $\mathbf{c}$, and matrices by bold capital Roman letters, e.g., $\mathbf{W}$. Additionally, subscripted bold letters denote specific vectors or matrices such as $\mathbf{k}_m$. Moreover, scalar quantities, such as $\lambda_k$, are written without bold. We denote the upper limits of scalar quantities by capitalized scalars, for example, $1 \leq n \leq N$.

## 4. Our Model

In this section, we present the Graph Hawkes Neural Network (GHNN) for modeling sequences of discrete large-scale multi-relational graphs in continuous time. The GHNN consists of the following two major modules:

- A neighborhood aggregation module for capturing the information from concurrent events that happened at the same timestamp.
- A Graph Hawkes Process for modeling the occurrence of a future fact where we use a recurrent neural network to learn this temporal point process.

We take the temporal knowledge graph as an example and show how our model deals with the link prediction task and the time prediction task. Besides, GHNN also learns latent representations specified for entities and predicates. In the rest of this section, we first define the relevant historical event sequence for each inference task, which is the input of GHNN, and then provide details on the proposed modules in GHNN.

## 4.1 Relevant Historical Event Sequences

In this work, we consider a temporal knowledge graph $\mathcal{G}$ as a sequence of graph slices $\{\mathcal{G}_1, \mathcal{G}_2, ...., \mathcal{G}_T\}$, where $\mathcal{G}_t = \{(e_s, e_p, e_o, t) \in \mathcal{G}\}$ denote a graph slice that consists of facts that occurred at the timestamp $t$. Additionally, inspired by [Jin et al., 2019], we assume that concurrent events belonging to the same graph slice, which means that they occurred at the same timestamp, are conditionally independent to each other given the past observed graph slices. Thus, for predicting a missing object entity of an object prediction query $(e_{s_i}, e_{p_i}, ?, t_i)$, we evaluate the conditional probability $\mathbb{P}(e_o|e_{s_i}, e_{p_i}, t_i, \mathcal{G}_{t_{i-1}}, \mathcal{G}_{t_{i-2}}, ..., \mathcal{G}_1)$ of all object entity candidates. To simplify the model complexity in this work, we assume that the conditional probability that an object entity forms a link with a given subject entity $e_{s_i}$ with respect to a predicate $e_{p_i}$ at a timestamp $t_i$ directly depends on past events that include $e_{s_i}$ and $e_{p_i}$. We define these events as the relevant historical event sequence $e_i^{h,sp}$ for predicting the missing object entity $e_{o_i}$:

$$e_i^{h,sp} = \{ \bigcup_{0 \leq t_j < t_i} (e_{s_i}, e_{p_i}, \mathbf{O}_{t_j}(e_{s_i}, e_{p_i}), t_j) \} \tag{2}$$

where $\mathbf{O}_{t_j}(e_{s_i}, e_{p_i})$ is a set of object entities that formed a link with the subject entity $e_{s_i}$ under the predicate $e_{p_i}$ at a timestamp $t_j$ ($0 \leq t_j < t_i$). Thus, we can rewrite the conditional probability of an object entity candidate $e_o$ given a query $(e_{s_i}, e_{p_i}, ?, t_i)$ and past graph slices, i.e., from $1^{st}$ to $(i-1)^{th}$, into the following form:

$$\mathbb{P}(e_o|e_{s_i}, e_{p_i}, t_i, \mathcal{G}_{t_{i-1}}, \mathcal{G}_{t_{i-2}}, ...., \mathcal{G}_1) = \mathbb{P}(e_o|e_{s_i}, e_{p_i}, t_i, e_i^{h,sp}). \tag{3}$$

To capture the impact of other past events that have different subject entity or predicate than the query has, we use a shared latent representation for an entity that appears in different quadruples. For each observed event in the training set, two entities involved in the event propagate information from the neighborhood of one entity to the other entity. Thus, after training, the model is also able to capture dynamics between multi-hop neighbors with various relations.

Similarly, we define a relevant historical event sequence $e_i^{h,op}$ for predicting the missing subject entity $e_{s_i}$ given a subject prediction query $(?, e_{p_i}, e_{o_i}, t_i)$. For the time prediction task, we assume that the time of the next occurrence of an event $(e_{s_i}, e_{p_i}, e_{o_i})$ is directly dependent on past events that include either $(e_{s_i}, e_{p_i})$ or $(e_{o_i}, e_{p_i})$. This gives the conditional probability density function at a timestamp $t$ given a query $(e_{s_i}, e_{p_i}, e_{o_i}, ?)$ and past graph slices with the following form:

$$p(t|e_{s_i}, e_{o_i}, e_{p_i}, \mathcal{G}_{t_{i-1}}, \mathcal{G}_{t_{i-2}}, ...., \mathcal{G}_1) = p(t|e_{s_i}, e_{o_i}, e_{p_i}, e_i^{h,sp}, e_i^{h,op}). \tag{4}$$

## 4.2 Neighborhood Aggregation

Because a subject entity can form links with multiple object entities within the same time slice, we use a mean aggregation module [Hamilton et al., 2017] to extract neighborhood information from concurrent events of a relevant historical event sequence. For predicting the missing object entity in an object prediction query $(e_{s_i}, e_{p_i}, ?, t_i)$, this module takes the element-wise mean of the embedding vectors of object entities in $\mathbf{O}_{t_j}(e_{s_i}, e_{p_i})$:

$$g(\mathbf{O}_{t_j}(e_{s_i}, e_{p_i})) = \frac{1}{|\mathbf{O}_{t_j}(e_{s_i}, e_{p_i})|} \sum_{e_o \in O_{t_j}(e_{s_i}, e_{p_i})} \mathbf{e}_o \tag{5}$$

where we denote the mean aggregation of embeddings of the neighboring object entities as $g(\mathbf{O}_{t_j}(e_{s_i}, e_{p_i}))$.

## 4.3 The Graph Hawkes Process

The time span between events often has significant implications on the underlying intricate temporal dependencies. Therefore, we model time as a random variable and deploy the Hawkes process on temporal knowledge graphs to capture the underlying dynamics. We call this the Graph Hawkes Process. In contrast to the classic Hawkes process with a parametric form, we use a recurrent neural network to estimate the intensity function $\lambda_k$ of the graph Hawkes process. Traditionally, recurrent neural networks are employed to sequential data with evenly spaced intervals. However, events in a temporal KG are randomly distributed in the continuous time space. Thus, inspired by the neural Hawkes process [Mei and Eisner, 2017] we use a continuous-time LSTM with an explicit time-dependent hidden state, where the hidden state is instantaneously updated with each event occurrence and also continuously evolves, as time elapses between two neighbored events. Specifically, given an object prediction query $(e_{s_i}, e_{p_i}, ?, t_i)$ and its relevant historical event sequence $e_i^{h,sp}$, we define the intensity function of an object candidate $e_o$ as follows:

$$\lambda(e_o|e_{s_i}, e_{p_i}, t_i, e_i^{h,sp}) = f(\mathbf{W}_\lambda(\mathbf{e}_{s_i} \oplus \mathbf{h}(e_o, e_{s_i}, e_{p_i}, t_i, e_i^{h,sp}) \oplus \mathbf{e}_{p_i}) \cdot \mathbf{e}_o) \tag{6}$$

where $\mathbf{e}_{s_i}, \mathbf{e}_{p_i}, \mathbf{e}_o \in \mathbb{R}^r$ are embedding vectors of the subject $e_{s_i}$, predicate $e_{p_i}$ and object $e_{o_i}$ of the event $e_i$, $\mathbf{h}(e_o, e_{s_i}, e_{p_i}, t_i, e_i^{h,sp}) \in \mathbb{R}^d$ denotes the hidden state of a continuous-time recurrent neural network that takes $e_i^{h,sp}$ as input and summarizes information of the relevant historical event sequence, and $\oplus$ represents the concatenation operator. $r$ and $d$ denote the rank of embeddings and the number of hidden dimensions, respectively. $\mathbf{W}_\lambda$ is a weight matrix which convert the dimensionality of the concatenation from $2r + d$ to $r$ so that we can form a dot-product between the concatenation and the embedding of the object candidate $e_o$. This captures the compatibility between $e_{s_i}$ and $e_o$ considering previous events they have been involved in.

Besides, to ensure that all elements of the intensity vector $\lambda(e_o|e_{s_i}, e_{p_i}, t_i, e_i^{h,sp})$ are strictly positive definite, we use the scaled softplus function as the activation function of the recurrent neural network, which is defined as:

$$f(x) = s \log(1 + \exp(x/s)). \tag{7}$$

All output values of the scaled softplus function are strictly positive definite and approach the corresponding outputs of the ReLU function as the scale parameter $s > 0$ approaches zero.

To let $\mathbf{h}(e_o, e_{s_i}, e_{p_i}, t_i, e_i^{h,sp})$ learn complex dependencies on the number, order and timing of the historical sequence $e_i^{h,sp}$, we adopt the continuous-time Long Short-Term Memory (cLSTM) [Mei and Eisner, 2017] since discrete-time approaches may fail to model the change of hidden states between two events when the time interval between them is considerable. We list some core functions in the following, the complete algorithm of a cLSTM cell is provided in Appendix B.

$$\mathbf{k}_m(e_{s_i}, e_{p_i}, e_i^{h,sp}) = g(\mathbf{O}_{t_m}(e_{s_i}, e_{p_i})) \oplus \mathbf{e}_{s_i} \oplus \mathbf{e}_{p_i} \tag{8}$$

$$\mathbf{c}(t) = \bar{\mathbf{c}}_{m+1} + (\mathbf{c}_{m+1} - \bar{\mathbf{c}}_{m+1}) \exp(-\boldsymbol{\delta}_{m+1}(t - t_m)) \tag{9}$$

$$\mathbf{h}(e_{s_i}, e_{p_i}, e_{o_i}, t, e_i^{h,sp}) = \mathbf{e}_{o_i} \cdot \tanh(\mathbf{c}(t)) \quad \text{for } t \in (t_m, t_{m+1}] \tag{10}$$

For capturing cumulative knowledge in the historical event sequence, the vector $\mathbf{k}_m(e_{s_i}, e_{p_i}, e_i^{h,sp})$ concatenates the neighborhood aggregation based on $\mathbf{O}_{t_m}(e_{s_i}, e_{p_i})$ with the embedding vector of the corresponding subject and predicate as the input of the cLSTM. Equations 9 and 10 make the memory cell vector $\mathbf{c}(t)$ discontinuously jump to a initial cell state $\mathbf{c}_{m+1}$ at each update of the cLSTM, and then continuously drift toward a target cell state $\bar{\mathbf{c}}_{m+1}$, which in turn controls the hidden state vector $\mathbf{h}(e_{s_i}, e_{p_i}, e_{o_i}, t, e_i^{h,sp})$ as well as the intensity function. The term $\mathbf{c}_{m+1} - \bar{\mathbf{c}}_{m+1}$ is related to the degree to which the past events influence the current events. The influence on each element of $\mathbf{c}(t)$ could be either excitatory or inhibitory, depending on the sign of the corresponding element of the decaying vector $\boldsymbol{\delta}_{m+1}$. Thus, the hidden state vector reflects how the system's expectations about the next occurrence of a specific event changes as time elapses and models structural and temporal coherence in the given tKG.

## 4.4 Inference and Parameter Learning

In this section, we will provide details about how the GHNN perform link prediction task and time prediction task. Besides, we will introduce the training procedure of the GHNN.

**Link prediction** Given an object prediction query $(e_{s_i}, e_{p_i}, ?, t_i)$ and its relevant historical event sequence $e_i^{h,sp}$, we derive the conditional density function of an object candidate $e_o$ from Equation 1, which gives the following equation,

$$p(e_o | e_{s_i}, e_{p_i}, t_i, e_i^{h,sp}) = \lambda(e_o | e_{s_i}, e_{p_i}, t_i, e_i^{h,sp}) \exp(- \int_{t_L}^{t_i} \lambda_{surv}(e_{s_i}, e_{p_i}, \tau) \, d\tau) \tag{11}$$

where $t_L$ denotes the timestamp of the most recent event in $e_i^{h,sp}$, and the integral represents the survival term [Daley and Vere-Jones, 2007] of all possible events $\{e_{s_i}, e_{p_i}, e_o = j\}_{j=1}^{N_e}$ with regarding to the given subject entity $e_{s_i}$ and the predicate $e_{p_i}$, which is defined as:

$$\lambda_{surv}(e_{s_i}, e_{p_i}, t) = \sum_{e_o=1}^{N_e} \lambda(e_{s_i}, e_{p_i}, e_o, t). \tag{12}$$

As shown in Equation 11, all object candidates share the same survival term $\lambda_{surv}(e_{s_i}, e_{p_i}, t)$ and the same value of $t_L$. Thus, at inference time, instead of comparing the conditional density function of each object candidate $e_o$, we can directly compare their intensity function $\lambda(e_o|e_{s_i}, e_{p_i}, t_i, e_i^{h,sp})$ to avoid the computationally expensive evaluation of the integrals.

**Time prediction** For the time prediction task, given an event $(e_{s_i}, e_{p_i}, e_{o_i})$, we aim to predict the expected time instance of its next occurrence based on observed events. Since we have full information about the involving subject entity and the object entity, we can utilize both $e_i^{h,sp}$ and $e_i^{h,op}$. Hence, the intensity that an event type $(e_{s_i}, e_{p_i}, e_{o_i})$ occurs at a future time $t$ is defined as follows:

$$
\begin{aligned}
\lambda(t|e_{s_i}, e_{p_i}, e_{o_i}, e_i^{h,sp}, e_i^{h,op}) = & f(\mathbf{W}_\lambda(\mathbf{e}_{s_i} \oplus \mathbf{h}(e_{o_i}, e_{s_i}, e_{p_i}, t_i, e_i^{h,sp}) \oplus \mathbf{e}_{p_i}) \cdot \mathbf{e}_{o_i}) \\
& + f(\mathbf{W}_\lambda(\mathbf{e}_{o_i} \oplus \mathbf{h}(e_{s_i}, e_{o_i}, e_{p_i}, t_i, e_i^{h,op}) \oplus \mathbf{e}_{p_i}) \cdot \mathbf{e}_{s_i}).
\end{aligned}
\tag{13}
$$

In the literature, the Hawkes process predicts when the next event will happen without regarding the event type. In contrast, our task here is to predict the time instance of the next occurrence of the given event type $(e_{s_i}, e_{p_i}, e_{o_i})$. Thus, we use a Hawkes process with a single event type to perform the time prediction[1]. This gives the corresponding conditional density function,

$$
\begin{aligned}
p(t|e_{s_i}, e_{p_i}, e_{o_i}, e_i^{h,sp}, e_i^{h,op}) = & \\
\lambda(t|e_{s_i}, e_{p_i}, e_{o_i}, e_i^{h,sp}, e_i^{h,op}) & \exp(-\int_{t_L}^t \lambda(\tau|e_{s_i}, e_{p_i}, e_{o_i}, e_i^{h,sp}, e_i^{h,op}) \, d\tau).
\end{aligned}
\tag{14}
$$

Accordingly, the expectation of the next event time is computed by:

$$
\hat{t}_i = \int_{t_L}^\infty t \cdot p(t|e_{s_i}, e_{p_i}, e_{o_i}, e_i^{h,sp}, e_i^{h,op}) \, dt
\tag{15}
$$

where the integrals in Equation 14 and 15 are estimated by the trapezoidal rule [Atkinson, 2008].

**Parameter learning** Because the link prediction can be viewed as a multi-class classification task, where each class corresponds to an entity candidate, we use the cross-entropy loss for learning the link prediction:

$$
\mathcal{L}_{\text{link}}^{\text{sp}} = -\sum_{i=1}^N \sum_{c=1}^{N_e} y_c \log(p(e_{o_i} = c|e_{s_i}, e_{p_i}, t_i, e_i^{h,sp}))
\tag{16}
$$

$$
\mathcal{L}_{\text{link}}^{\text{op}} = -\sum_{i=1}^N \sum_{c=1}^{N_e} y_c \log(p(e_{s_i} = c|e_{o_i}, e_{p_i}, t_i, e_i^{h,op}))
\tag{17}
$$

where $\mathcal{L}_{\text{link}}^{\text{sp}}$ is the loss of object prediction given the query $(e_{s_i}, e_{p_i}, ?, t_i)$ and $\mathcal{L}_{\text{link}}^{\text{op}}$ is the loss of subject prediction given the query $(?, e_{p_i}, e_{o_i}, t_i)$, and $y_c$ is a binary indicator of whether class label $c$ is the correct classification for predicting $e_{o_i}$ and $e_{s_i}$. In addition, we use the

---

1. It can be easily derived from the Equation 1 that the integration of the density function of the Hawkes process with a single event type is one.

mean square error as the time prediction loss $\mathcal{L}_{\text{time}} = \sum_{i=1}^{N}(t_i - \hat{t}_i)^2$. Hence, the total loss is the sum of the time prediction loss and the link prediction loss:

$$\mathcal{L} = \mathcal{L}_{\text{link}}^{\text{sp}} + \mathcal{L}_{\text{link}}^{\text{op}} + \nu\mathcal{L}_{\text{time}}. \tag{18}$$

Additionally, we balance the time prediction loss and the link prediction loss by scaling the former using a hyperparameter $\nu$. The gradient backpropagation is automatically done by PyTorch [Paszke et al., 2019]. The learning algorithm of the GHNN is described in the Appendix D. Also, we illustrated the architecture of the GHNN in Appendix E.

## 5. Experiments

### 5.1 Experimental Setup

**Datasets**  Global Database of Events, Language, and Tone (GDELT) [Leetaru and Schrodt, 2013] dataset and Integrated Crisis Early Warning System (ICEWS) [Boschee et al., 2015] dataset have been drawing attention in the community as suitable examples of tKGs [Schein et al., 2016]. The GDELT dataset is an initiative to construct a database of all the events across the globe, connecting people, organizations, and news sources. We use a subset of the GDELT dataset, which contains events occurring from January 1, 2018 to January 31, 2018. The ICEWS dataset contains information about political events with specific time annotations, e.g. (Ban Ki-moon, Secretary-General of, the United Nations, 2007-01-01). We apply our model on a subset ICEWS14 of the ICEWS dataset, which contains events occurring in 2014. We compare our approach and baseline methods by performing the link prediction task as well as the time prediction task on the GDELT dataset and the ICEWS14 dataset. Appendix F provides detailed statistics about the datasets.

**Implementation details of the GHNN**  By training the GHNN, we set the maximal length of historical event sequences to be 10, the size of embeddings of entities/predicates to be 200, and the learning rate to be 0.001. The model is trained using the Adam optimizer. We set the weight decay rate to be 0.00001, and the batch size to be 1024. The above configurations were used for all experiments that were done on GeForce GTX 1080 Ti.

**Evaluation metrics**  In the literature, there are different metrics for evaluating the results of link prediction on semantic KGs. The mean reciprocal rank (MRR) is one of those commonly used evaluation metrics, where we remove an entity (subject or object) of a test triplet $(e_{s_i}, e_{p_i}, e_{o_i})$, replace it with by all entities that can potentially be the missing entity, find the rank of the actual missing entity, and then take the reciprocal value. Besides, some researchers use Hits@$K$ to evaluate the model's performance, which is the percentage that the actual missing entity is ranked in the top $K$. However, these metrics can be flawed when some corrupted triplets end up being valid ones, from the training set for instance. In this case, those may be ranked above the actual missing entity, but this should not be seen as an error because both triplets are true. Bordes et al. [2013] suggested removing from the list of corrupted triplets all the triplets that appear either in the training, validation, or test set except the test triplet of interest, which ensures that all corrupted triplets do not belong to the dataset. Trivedi et al. [2017] and Jin et al. [2019] used the ranking technique described in [Bordes et al., 2013] for evaluating the link prediction on temporal KGs. For

example, there is a test quadruple (Barack Obama, visit, India, Jan. 25, 2015), and we perform the object prediction (Barack Obama, visit, ?, Jan. 25, 2015). Besides, we observe (Barack Obama, visit, Germany, Jan. 18, 2013) in the training set. According to the ranking technique described in [Bordes et al., 2013], (Barack Obama, visit, Germany, Jan. 25, 2015) is considered to be valid since the triplet (Barack Obama, visit, Germany) appears in the training set. However, we think this ranking technique is not appropriate for temporal KGs since the triplet (Barack Obama, visit, Germany) is only temporally valid on Jan. 18, 2013 but not on Jan. 25, 2015. Therefore, we define a new ranking procedure. For the object prediction (Barack Obama, visit, ?, Jan. 25, 2015), instead of removing from the list of corrupted events all the events that appear either in the training, validation or test set, we only filter from the list all the events that occur on Jan. 25, 2015. This ensures that the triplet (Barack Obama, visit, Germany) is still considered as invalid on Jan. 25, 2015. Additionally, since all object candidates are ranked by their scores, some entities may have identical scores. In this case, most papers give the highest rank of all entities, leading that the rank may be incredibly high even if the estimator makes a trivial prediction, i.e. giving identical scores to all entity candidates. For a fair evaluation, we give a mean rank to entities that have same scores. For the time prediction task, Trivedi et al. [2017] used the mean absolute error (MAE) between the predicted time and the ground-truth to evaluate the experiment results. However, a small part of bad predictions may lead to a high MAE although the majority of predictions has good quality. Thus, we propose the continuous Hits@$k$ (cHits@$k$) for the time prediction task where cHits@$k$ is defined as the ratio of data samples whose MAE is smaller than $k$.

**Baseline methods**   For the link prediction task, we compare the performance of our model with several state-of-the-art methods for tKGs, including TTransE [Leblay and Chekol, 2018], TA-TransE/Distmult [García-Durán et al., 2018], Know-Evolve [Trivedi et al., 2017], and RE-Net [Jin et al., 2019]. For the time prediction task, we compare our model only with LiTSEE [Xu et al., 2019] and Know-Evolve since only these two models are capable of performing the time prediction task on tKGs to the best of our knowledge. We provide the implemetation details of these baselines in Appendix G.

Table 1: Link prediction results: MRR (%) and Hits@1/3/10 (%).

| Datasets | ICEWS14 - filtered | | | | GDELT - filtered | | | |
|---|---|---|---|---|---|---|---|---|
| Metrics | MRR | Hits@1 | Hits@3 | Hits@10 | MRR | Hits@1 | Hits@3 | Hits@10 |
| T-TransE | 7.15 | 1.39 | 6.91 | 18.93 | 5.45 | 0.44 | 4.89 | 15.10 |
| TA-TransE | 11.35 | 0.00 | 15.23 | 34.25 | 9.57 | 0.00 | 12.51 | 27.91 |
| TA-Dismult | 10.73 | 4.86 | 10.86 | 22.52 | 10.28 | 4.87 | 10.29 | 20.43 |
| LiTSEE | 6.45 | 0.00 | 7.00 | 19.40 | 6.64 | 0.00 | 8.10 | 18.72 |
| Know-Evolve | 1.42 | 1.35 | 1.37 | 1.43 | 2.43 | 2.33 | 2.35 | 2.41 |
| RE-Net | 28.56 | 18.74 | 31.49 | **48.54** | 22.24 | 14.24 | 23.95 | 38.21 |
| GHNN | **28.71** | **19.82** | **31.59** | 46.47 | **23.55** | **15.66** | **25.51** | **38.92** |

## 5.2 Performance Comparison on Temporal Knowledge Graphs

**Link prediction results** Table 1 summarizes link prediction performance comparison on the ICEWS14 and GDELT datasets. GHNN gives on-par results with RE-Net and outperforms all other baseline models on these datasets considering MRR, Hits@1/3/10. Know-Evolve shows poor performance due to its limited capability of dealing with concurrent events. Additionally, our model beats RE-Net because they only consider the temporal order between events. In comparison, GHNN explicitly encodes time information into the intensity function, which improves the expressivity of our model. The results indicate that the Graph Hawkes Process substantially enhances the performance of reasoning on tKGs.

**Time prediction results** Table 2 demonstrates that GHNN performs significantly better than LiTSEE for the time prediction task on both the ICEWS14 dataset and the GDELT dataset. This result shows the superiority of the GHNN compared to methods that model tKGs by merely adding a temporal component into entity embeddings. Furthermore, Know-Evolve has good results on the ICEWS14 dataset due to its simplest ground-truth distribution, which is shown in Appendix H. In particular, according to the settings of Know-Evolve, most ground-truth values for the time prediction task are exactly zero. The reason is that, for a ground-truth quadruple $(s, p, o, t)$, Know-Evolve defines the ground-truth value for time prediction as the difference between the timestamp $t$ and the most recent timestamp $t'$ when either the subject entity $s$ or the object entity $o$ was involved in an event. However, they do not consider concurrent events. For example, we have events $e_1 = (s, p, o_1, t_1)$ and $e_2 = (s, p, o_2, t_1)$. After $e_1$, $t'$ becomes $t_1$ (most recent event time of subject $s$), and thus the ground-truth value of $e_2$ for the time prediction task is 0.

Table 2: Time prediction results: MAE and cHits@1/3/10 (%). $^+$ indicates results in this row were taken from [Trivedi et al., 2017].

| Datsets | ICEWS14 | | | GDELT | | |
|---|---|---|---|---|---|---|
| Metrics | MAE (**days**) | cHits@1 | cHits@10 | MAE (**hours**) | cHits@1 | cHits@10 |
| Know-Evolve$^+$ | **1.78** | - | - | 110.8 | - | - |
| LiTSEE | 108.00 | - | 25.10 | 303.78 | - | 0.00 |
| GHNN | 6.10 | 68.73 | 90.80 | **7.18** | 58.79 | 89.38 |

## 6. Conclusion

We presented the Graph Hawkes Neural Network, a novel neural architecture for forecasting on temporal knowledge graphs. To model the temporal dynamics of tKGs, we proposed the Graph Hawkes Process, a multivariate point process model of streams of timestamped events, that can capture underlying dynamics across facts. The model parameters are learned via a continuous-time recurrent neural network, which is able to estimate the probability of events at an arbitrary instance in the future. We test our model on two temporal knowledge graphs,

where experimental results demonstrate that our approach outperforms the state-of-the-art methods on link prediction and time prediction over tKGs.

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
