# OpenReview forum: "Graph Hawkes Neural Network for Forecasting on Temporal Knowledge Graphs"
_AKBC.ws/2020/Conference — AKBC 2020_

### Official Review · AnonReviewer2 · 2020-03-26
**The paper needs more improvement.**

**Rating:** 5
**Confidence:** 5

**Review:**

This work proposes a graph Hawkes Neural Network for event and time prediction on temporal knowledge graphs. Overall, the paper is nicely written and easy to follow.
However, this paper needs more improvement. Also, the proposed method needs to compare other baseline methods and is somewhat limited. Importantly, the analysis of the proposed method is missing.

Major comments
- About equation (3), the narrative looks wrong and I don't understand why this is equivalent to the conditional probability P(e_o|e_si, e_pi, t_i, g,...). There should be explanations on P(e_si, e_pi| g,...).
- The proposed method only considers limited information from historical event sequences. For example, in equation (5) do not include multi-relational entities other than events under the same predicate. Furthermore, it does not capture multi-hop neighbors explicitly.
- What is the major advantage of using cLSTM? It would be better to compare with LSTM, Time-LSTM [1], and Time-aware LSTM [2]
- What is the definition of lambda_sub in equation (14)?
- The definition of t_L is missing. Also I don't understand how to predict time using equation (16).
- Furthermore, if the paper assumes that there is only one single event type, then given all the events, the method should predict the same next event time?
- About the experiments, it would be better to include static approaches such as ConvE [3], RotatE [4].
- Also if the method uses LSTM instead of cLSTM, then what will be the result? This is necessary to show the effectiveness of cLSTM.

Minor comments
- The example in the introduction is not appropriate. The paper talks about the price of the oil (attribute of a node), but the proposed method does not predict attribute value.

[1] What to Do Next: Modeling User Behaviors by Time-LSTM
[2] Patient Subtyping via Time-Aware LSTM Networks
[3] Convolutional 2d knowledge graph embeddings
[4] Rotate: Knowledge graph embedding by relational rotation in complex space

---

> ### Author Response · Authors · 2020-04-09
> **Response to reviewer2 [2/2]**
>
>
> Q6: Furthermore, if the paper assumes that there is only one single event type, then given all the events, the method should predict the same next event time?
>
> A6: Sorry for the confusion caused by the unclear formulation in our paper, and the answer is no.  Even for a single event type $e_i$, since its historical event sequence changes over time, this event type has different next event time when querying at different timestamps.
>
> Moreover, in our paper, we don't mean that we ignore the labels for events and assume there is only one single event type in the whole timeline. For a given event type $e_i = (e_{s_i}, e_{p_i}, e_{o_i})$, we still consider all event types in the past ($t$ < $t_L$). Thus, according to Equation 15, different events would have different intensity functions $\lambda(t|e_{s_i}, e_{p_i}, e_{o_i}, e^{h,sp}_i, e^{h,op}_i)$, and therefore different next event time.
> ---------------------------------------------------------------------------------------------------------------------------------------------------------
> Q7: About the experiments, it would be better to include static approaches such as ConvE [3], RotatE [4].
>
> A7: Thanks for the suggestion of strengthening our experiments. The main reason why we didn’t include these static approaches is that these approaches require to use filtered metrics for semantic knowledge graphs as defined in [1], which is different from our newly defined filtered metrics in Section 5.1 of our manuscript.
>
> For evaluating static approaches, we compress temporal knowledge graphs into a static, cumulative graph by ignoring edge timestamps. We have added Table 6 in the appendix of our paper to show the raw results of these models. Here, we briefly summarize the results on the GDELT dataset as follows. More detailed results can be found in Table 6 of the updated draft.
>
>
> Models	  | MRR(%)| Hits@1(%)|Hits@3(%)|Hits@10(%)
> —————————————————————————--
> ComplEx        9.84            5.17             9.58            18.23
> RotatE            3.62            0.52             2.26             8.37
> ConvE            18.37          11.29           19.36           32.13
> —————————————————————————--
> T-TransE        5.53            0.46              4.97            15.37
> TA-DistMult  10.34          4.44              10.44           21.63
> KnowEvolve  0.11            0.00              0.02             0.10
> Re-Net           19.60          12.03            20.56          33.89
> —————————————————————————--
> GHNN            22.89          14.88            24.90          38.48
>
>
> We can see that GHNN outperforms these seven baselines by a large margin. Static methods show good results, but they underperform our approach. We conjecture that the static approaches are not specifically designed for temporal knowledge graphs. Thus, they are not capable of handling temporal factors, which leads to the degradation of their performances.
>
> [1] Antoine Bordes, Nicolas Usunier, Alberto Garcia-Duran, Jason Weston, and Oksana Yakhnenko. Translating embeddings for modeling multi-relational data. In Advances in neural information processing systems, pages 2787–2795, 2013.
> ---------------------------------------------------------------------------------------------------------------------------------------------------------
> Q8: Also if the method uses LSTM instead of cLSTM, then what will be the result? This is necessary to show the effectiveness of cLSTM.
>
> A8: Thank you for making a good point to strengthen our experiment. However, if the method uses LSTM, it is not able to perform the time prediction task, which is the necessity of using cLSTM.
> ---------------------------------------------------------------------------------------------------------------------------------------------------------
> Q9: The example in the introduction is not appropriate. The paper talks about the price of the oil (attribute of a node), but the proposed method does not predict attribute value.
>
> A9: Thank you for pointing it out. We have made another example and updated the introduction accordingly.

---

> ### Author Response · Authors · 2020-04-09
> **Response to reviewer2 [1/2]**
>
> Thank you very much for your helpful feedback and valuable comments! We tried to resolve the issues and incorporate the comments in our updated draft.
> ---------------------------------------------------------------------------------------------------------------------------------------------------------
> Q1: About equation (3), the narrative looks wrong and I don't understand why this is equivalent to the conditional probability P(e_o|e_si, e_pi, t_i, g,...). There should be explanations on P(e_si, e_pi| g,...).
>
> A1: Thank you for pointing it out. We agree that our narrative about Equation 3 is imprecise. We have updated the writing in Section 4.1.
> ---------------------------------------------------------------------------------------------------------------------------------------------------------
> Q2: The proposed method only considers limited information from historical event sequences. For example, in equation (5) do not include multi-relational entities other than events under the same predicate. Furthermore, it does not capture multi-hop neighbors explicitly.
>
> A2: Thank you for the comment. We agree that we do not explicitly capture multi-hop neighbors. However, since each entity has a unique embedding, for each observed quadruple, two entities involved in the event would propagate information from the neighborhood of one entity to the other entity. Thus, after training, the model is also able to capture dynamics between multi-hop neighbors with various relations.
>
> Moreover, as shown in Appendix K of our paper, we do explicitly consider multi-relational neighbors for subject-predicate (object-predicate) pairs that didn't interact in the past. More details can be found in Appendix K.
> ---------------------------------------------------------------------------------------------------------------------------------------------------------
> Q3: What is the major advantage of using cLSTM? It would be better to compare with LSTM, Time-LSTM [1], and Time-aware LSTM [2].
>
> A3: Thank you for the suggestion! The main reasons why we didn’t include a detailed discussion on these related models are: (1) we cannot adapt the Hawkes process to LSTM, Time-LSTM, and Time-aware LSTM. In particular, they are not able to simulate the assumption that the influence of past events exponentially decays with time. (2) Nither LSTM nor Time-LSTM and Time-aware LSTM can compute the conditional density function of event time $p(t|e_{s_i}, e_{p_i}, e_{o_i}, e^{h,sp}_i, e^{h,op}_i)$. Thus, these models cannot perform the time prediction task.
> ---------------------------------------------------------------------------------------------------------------------------------------------------------
> Q4: What is the definition of lambda_sub in equation (14)?
>
> A4: Thanks for pointing out and sorry for the confusion. We have fixed the typo about Equation 14. It should be $\lambda (e_{s_i}, e_{p_i}, e_{o}, t)$, which means the intensity of a quadruple $(e_{s_i}, e_{p_i}, e_o, t)$. And the survival term $\lambda_{surv}(e_{s_i}, e_{p_i}, t)$ for object prediction represents the intensities of all possible events $\{e_{s_i}, e_{p_i}, e_o = j\}_{j=1}^{N_e}$ regarding the given subject entity $e_{s_i}$ and the predicate $e_{p_i}$. We see that all object candidates share the same survival term $\lambda_{surv}(e_{s_i}, e_{p_i}, t)$ and the same value of $t_L$. The corresponding conditional density functions also have the same integral in Equation 13. Thus, instead of comparing  the conditional density function of each object candidate $e_o$, we can directly compare their intensity function $\lambda(e_{o} | e_{s_i}, e_{p_i}, t_i, e^{h, sp}_i)$ to avoid the computationally expensive integral calculations.
> ---------------------------------------------------------------------------------------------------------------------------------------------------------
> Q5: The definition of t_L is missing. Also I don't understand how to predict time using equation (16).
>
> A5: Sorry for the confusion. Because $t_L$ firstly appears in Equation 13, we have defined it in the paragraph below this equation. In general, $t_L$ denotes the timestamp of the most recent event in historical event sequences. In Equation 16, it means the timestamp of the most recent event considering both $e^{h, sp}_i$ and $e^{h, op}_i$.
>
> To predict the next event time, We compute the expected value using $\hat t_i = \int_{t_L}^{\infty} t \cdot p(t|e_{s_i}, e_{p_i}, e_{o_i}, e^{h,sp}_i, e^{h,op}_i)\;dt$ , where $p(t|e_{s_i}, e_{p_i}, e_{o_i}, e^{h,sp}_i, e^{h,op}_i)$ is defined in Equation 16. We have added the above equation in Section 4.4 of our manuscript to explain the time prediction procedure clearly.

---

### Official Review · AnonReviewer1 · 2020-03-29

**Rating:** 7
**Confidence:** 3

**Review:**

This paper introduces a new model based on the Hawkes process to model the temporal knowledge base. The proposed method is based on previous work Mei & Eisen, 2017. The previous work target event prediction and time prediction, but in this paper, the task is subject/object prediction and time prediction. Furthermore, the author showed result improvement over strong baselines on several datasets.

Pros:
The paper is well written and the model is well motivated. The experiments show improvement with the proposed model compared to baselines.

Cons:
The results from different models are very close to each other. Is the proposed model significantly better than other baselines? Can you run the significance test on the results?

---

> ### Author Response · Authors · 2020-04-09
> **Response to reviewer1**
>
> Thank you for making an excellent point to strengthen our experiment. We agree that, for link prediction, the performance of our model is similar to Re-Net. But except the Re-Net, our model performs considerably better than other baselines, as shown in Tables 1 and 2 in our manuscript. Also, the key advantage of our model is that it can perform both the time prediction task and the link prediction task, while the Re-Net can only make link prediction.
>
> For the time prediction task, we compared our model with LiTSEE and Know-Evolve since only these two models are capable of performing the time prediction task on temporal knowledge graphs. Experimental results demonstrate that our model performs significantly better than LiTSEE on both the ICEWS14 dataset and the GDELT15/16 dataset. Furthermore, Know-Evolve has good results on the ICEWS14 dataset due to its simplest ground truth distribution. As shown in Appendix I, according to the faulty settings of Know-Evolve, most ground-truth values for the time prediction task are zeros. The reason is that, for a ground-truth quadruple $(s, p, o, t)$, the ground-truth value for the time prediction task is defined as the difference between the timestamp $t$ and the most recent timestamp $t'$ when either the subject entity $s$ or the object entity $o$ was involved in an event. However, they don’t consider concurrent events at the same timestamp, and thus t will become $t'$ after one event. For example, we have events $e_1 = (s, p, o_1, t_1), e_2 = (s, p, o_2, t_1)$. After $e_1$, $t'$ will become $t_1$, (most recent interacting timestamp of the subject $s$), and thus the ground-truth value of $e_2$ for the time prediction task will be $0$. Thus, compared to our model, Know-Evolve does not show its ability to make long-term predictions.

---

### Official Review · AnonReviewer3 · 2020-03-30
**Well formulated solution framework, excellent results, a few details missing**

**Rating:** 8
**Confidence:** 4

**Review:**

The paper addresses the problem of predicting links and time-stamps in a temporal knowledge graph, and proposes a novel neural Hawkes model that uses the continuous neural Hawkes formulation as its basis. Key assumption the authors make is that the object entity interacts (forms a temporal link) with a predicate p with a subject entity s only based on past links involving the same s and p. Thus, the history can be im
plicitly modeled by aggregating across all the objects. After this, the rest of the model quite closely follows that of Mei & Eisner model. The experiments conducted over GDELT and ICEWS14 datasets show that th
e proposed GHNN offers significantly better results than Know-Evolve, and is nearly as good as the closest RE-Net. Overall, the paper is well written and the model seems sound.

A few details are missing though:
1. They claim "NH approach is limited by the number of event types" but they haven't compared with it in any experiment
2. Knowevolve implementation as in Appendix I gives ~90% predictions when there is no actual change in time -- for both GDELT and ICEWS datasets. Yet the performance of KnowEvolve in time prediction on these two datasets is significantly different. Unclear what is going on here.
3. It is not known how much time-series information is available for each fact (not all subject/predicate pairs will be interacting in the previous slices).

---

> ### Author Response · Authors · 2020-04-09
> **Response to reviewer3 [2/2]**
>
>
> Q2: Knowevolve implementation as in Appendix I gives ~90% predictions when there is no actual change in time -- for both GDELT and ICEWS datasets. Yet the performance of KnowEvolve in time prediction on these two datasets is significantly different. Unclear what is going on here.
>
> A2: The results of KnowEvolve in time prediction on both GDELT and ICEWS datasets, which is the third row of Table 2 in our manuscript, are reported in the respective paper titled "Know-Evolve: Deep Temporal Reasoning for Dynamic Knowledge Graphs".
>
> Notably, we want to point out in Appendix I that the ground-truth value of time prediction is ill-defined in KnowEvolve. For a ground-truth quadruple $(s, p, o, t)$, the ground-truth value for the time prediction task is defined as the difference between the timestamp $t$ and the most recent timestamp $t'$ when either the subject entity $s$ or the object entity $o$ was involved in an event. However, they don’t consider concurrent events at the same timestamp, and thus $t$ will become $t'$ after one event. For example, we have events $e_1 = (s, p, {o_1}, t_1)$, $e_2 = (s, p, {o_2}, t_1)$. After $e_1$, $t'$ will become $t_1$, (most recent interacting timestamp of the subject $s$), and thus the ground-truth value of $e_2$ for the time prediction task will be $0$.
> ---------------------------------------------------------------------------------------------------------------------------------------------------------
> Q3: It is not known how much time-series information is available for each fact (not all subject/predicate pairs will be interacting in the previous slices).
>
> A3: Thanks for the great question. We add some histograms about the length of the history of each query in Appendix M of our paper. Here, we briefly summarize the results on the ICEWS14 dataset as follows. More detailed results can be found in the updated manuscript.
>
> object prediction given subject-predicate pairs $(e_{s_i}, e_{p_i}, ?, t_i)$:
> ————————————————————————————————
> length of past history: 0,          1,           2,             3,           4,         5,            >=6:
> number of pairs:         67034,  32531,   22394,   17337,  14380, 12435,   326568.
> ————————————————————————————————
> Taking subject-predicate pairs as an example, we see that more than 86% of them interacted at least once in the previous slices. In particular, more than 66% of them interacted more than five times in the past.

---

> ### Author Response · Authors · 2020-04-09
> **Response to reviewer3 [1/2]**
>
> Thank you very much for your helpful feedback and valuable comments! We appreciate the questions you raised regarding related work/baselines and analysis of available time-series information. We tried to resolve the issues and incorporate the comments in our updated draft.
> ---------------------------------------------------------------------------------------------------------------------------------------------------------
> Q1: They claim "NH approach is limited by the number of event types" but they haven't compared with it in any experiment.
>
> A1: Thank you for asking for additional comparisons to help validate our claims. The main reason why we didn’t include experiments on Neural Hawkes (NH) approach is they study event sequences and give each event type a one-hot encoding, while our work focuses on sequences of knowledge graphs and represent each event type as a semantic triple (subject, predicate, object). Thus, there are two issues if applying the neural Hawkes approach for temporal knowledge graph reasoning.
>
> First, the number of model parameters in the NH approach explodes on temporal knowledge graphs. Specifically, the number of learnable parameters of NH is about $N_{event} \cdot N_{hidden}$, where $N_{event}$ represents the number of events, and $N_{hidden}$ denotes the number of hidden units. Assuming the number of entities ($N_e$) and predicates ($N_p$) is fixed, the number of all possible event types on a temporal knowledge graph is of order $N_e^2 \cdot  N_p$. Notably, we should take all possible events into account instead of the actual events we observed in the training set because, for future prediction, there are new event types in the test set that we haven't seen in the past. Thus, taking the ICEWS14 dataset as an example, the potential number of event types ($N_{event}$) is $12498 \cdot  12498 \cdot  254 \approx 3.9 \cdot 10^{10}$.  Thus, the learnable parameters of the NH approach are at least thirty-nine billion, while we only have about five hundred thousand training samples. Therefore, it is not feasible to train the NH model on temporal knowledge graphs.
>
> Second, the NH approach is not able to make conditional inference on knowledge graphs. In particular, a classic task on knowledge graphs is predicting a missing object entity $(e_{s_i}, e_{p_i}, ?)$. However, the NH approach encodes each event/triplet with a one-hot vector. Thus, it is difficult to feed information about the given subject/predicate into the model.

---

### Author Response · Authors · 2020-04-09
**Paper revision**


We would like to thank the reviewers for their thoughtful comments and efforts towards
improving our manuscript. We have carefully revised the paper to address the comments from reviewers, and also added additional experiments to support our approach. The major updates of the draft are listed as follows:

- Adding experiments of other baseline models in Table 6 of the appendix, as suggested by reviewers.
- Adding figures about available time-series information of each fact in Section M of the appendix.
- Improved clarity of Sections 4 to reflect the questions raised by reviewers.

---

### Author Response · Authors · 2020-06-07
**Link to Supplementary Materials**

Please find the appendix under the following link https://drive.google.com/file/d/1h9WzNqp2r4IRxOfRehfQ99efzYq6Mrw1/view?usp=sharing .

---

### Decision · Program_Chairs · 2020-04-30

**Decision:**

Accept

**Comment:**

This paper propose Graph Hawkes Neural Networks (GHNN) which are suited for performing inference in temporal knowledge graphs. By combining the continuous modeling provided by cLSTM with strong assumptions about how events can impact one another, GHNN shows promising results compared strong baselines on the GDELT and ICEWS14 dataset.